# The Effect of Cutting Fluid on Machined Surface Integrity of Ultra-High-Strength Steel 45CrNiMoVA

**DOI:** 10.3390/ma16093331

**Published:** 2023-04-24

**Authors:** Yubin Wang, Yan Ren, Pei Yan, Siyu Li, Zhicheng Dai, Li Jiao, Bin Zhao, Siqin Pang, Xibin Wang

**Affiliations:** 1School of Mechanical Engineering, Beijing Institute of Technology, No. 5 Zhongguancun South Street, Haidian District, Beijing 100081, China; 2Beijing North Vehicle Group Corporation, No. 5 Wuli, Zhujiafen, Fengtai District, Beijing 100072, China; 3Key Laboratory of Fundamental Science for Advanced Machining, Beijing Institute of Technology, No. 5 Zhongguancun South Street, Haidian District, Beijing 100081, China

**Keywords:** cutting liquid, ultra-high-strength steel, surface quality, dynamic mechanical property, microstructure characterization

## Abstract

The surface integrity of ultra-high-strength steel has a significant influence on service performance, and cutting fluid plays an important role in maintaining surface integrity in production. In this paper, the surface integrity of ultra-high-strength steel 45CrNiMoVA was investigated under three cutting fluids: HY-103 (micro-emulsion), TRIM E709 (emulsion), and Vasco 7000 (micro-emulsion) from the aspects of cutting force, surface morphology, residual stress, micro hardness, microstructure, etc. The results showed that the changing trend of the cutting forces in three directions is HY-103 > Vasco 7000 > TRIM E709. The TRIM E709 contains the maximum lubricants, which reduce cutting force and Sa roughness, while the Vasco 7000 contains the minimum corrosive elements, which results in the least pitting. Both tangential and axial stresses under cutting fluid are tensile stresses. TRIM E709 and Vasco 7000 are reduced axially by 4.45% and 7.60% relative to HY-103, respectively. The grain refinement layer depths of HY-103, TRIM E709, and Vasco 7000 are 9 μm, 4 μm, and 8 μm, respectively, and TRIM E709 can induce recrystallized grains to grow along {001} of the sample cross section, which results from the lowest cooling rate. This work may provide an innovative control strategy for cutting fluid to improve surface integrity and service performance.

## 1. Introduction

Ultra-high-strength steel has excellent mechanical properties, and it has always been used as key high-load structural parts in the transmission systems of vehicles, ships, and aviation [1,2]. As these parts are subjected to a high load and harsh service environment, fatigue cracks usually originate from the surface, which seriously restricts the performance of service. Machined surface integrity, such as surface morphology, residual stress, and microstructure, directly determines the crack initiation on the surface and propagation in the subsurface. As one of the typical difficult-to-machine materials, ultra-high-strength steel always needs cutting fluid to reduce temperature and friction during material machining. Due to differences in composition and properties, cutting fluid has an important influence on surface integrity [3]. That is, crack initiation and propagation in the steel can be attributed to the surface integrity under cutting fluid [4].

Benefiting from its basic cooling and lubrication function, cutting fluid reduces temperature and friction, which improves cutting efficiency and surface integrity [5,6,7]. Simple cutting fluids were used in production in ancient times and then developed very slowly over a long period of time. It was only since the early 20th century that cutting fluids began to be used on a large scale and were developed with the advent of various machine tools. Different types and contents of cutting fluids were fabricated one after another, and their classification, compounds, recycling, and disposal were extensively studied [8,9,10]. Research on them has long focused on the basic properties of cooling, lubrication, cleaning, and rust prevention [11,12]. Open standards such as ASTM D2881-03-2009, JIS K2241-2000, and GB/T 6144-2010 also concentrate on evaluating the performance of cutting fluids and are used to guide the development and production of cutting fluids. At present, with the increasing awareness of environmental protection, new environmentally friendly cutting fluids and new application methods have been intensively researched [13,14,15,16,17].

Cutting fluid is divided into water-based cutting fluid and oil-based cutting fluid. Water-based cutting fluid has good cooling and cleaning properties. Because the specific heat capacity and thermal conductivity of water are much higher than those of oil, and water-based cutting fluid generally contains more surfactants, its washing ability is strong, making it easy to chip and clean dirt away. At the same time, the residual adsorption layer of oil-based cutting fluid on the workpiece surface is thick and dense, and the lubrication performance and rust prevention ability are better. In addition, the cost of direct use of oil-based cutting fluid is high, the cutting temperature is high, and it is easy to fog fire, which is harmful to the human body and environment. So, this paper mainly studies water-based cutting fluid.

In the development of precision and ultra-precision machining, surface integrity is more important than cutting efficiency, especially for critical high-load components such as torsion shafts and gears that serve in extreme operation conditions. However, the influence of cutting fluid on surface integrity and serviceability has not been systematically studied. In recent years, improper use of cutting fluid has resulted in numerous serious accidents due to poor surface integrity. Therefore, more and more attention has been paid to the influence of cutting fluid on surface integrity.

According to the current status of research on cutting fluids, the existing research is mainly focused on application methods and penetration ability and the effects on cutting performance, such as cutting force, tool wear, and chip formation [18,19,20]. However, the effects on machined surface integrity such as surface roughness, residual stress, and microhardness are less studied and will become an important research direction [21,22]. In the process of cutting, cutting fluids have a complex physicochemical response on the freshly machined surface due to the different composition and properties in the primary and third deformation zones (high temperature and high pressure zones). Yang et al. [11] developed a new synthetic water-based cutting fluid for titanium alloys, the polar groups of which were easy to adsorb on the alloy and formed a stable boundary lubricant film. The work found that the developed cutting fluid could decrease the cutting force and tool wear to obtain better surface quality. Srikant et al. [23] carried out hardness and corrosion tests on the AISI 1040 steel samples machined using the formulated fluids with vegetable-based alternatives and regular cutting fluid (with petroleum-based emulsifier). It was found that the fluid with 15% vegetable emulsifier had hardness similar to regular cutting fluid, but the fluid with 20% emulsifier had lesser corrosion compared to other fluids, and its chemical composition inhibits corrosion.

The composition of cutting fluid, especially the content of additives, has a significant influence on its physical and chemical properties, which are reflected in the machining properties and surface integrity [24,25,26]. Pereira et al. [27] analyzed the use of natural oil as a substitute for traditional canola oils in MQL and compared five different substitutes (namely, sunflower oil, high-oleic sunflower oil, castor oil, and ECO-350 recycled oil). The results indicate that the combination of low viscosity and high friction coefficient means an increase in tool life. In this regard, ECO-350 recycled oil exhibits feasible behavior, which can increase tool life by 30% compared to commercial canola oil.

There is a problem of suitability between the cutting fluid and the metal material being machined. On the one hand, different types of metal are treated with the same cutting fluid, and the surface integrity can be significantly different. Ruzzi et al. [28] aimed to analyze the surface finish and morphology of two nickel-based superalloys (Inconel 625 and Inconel 718) after grinding under semi-synthetic, environmentally friendly oil enriched with MLG (multilayer graphene) platelets via the MQL (minimum quantity lubrication) technique. The results for Inconel 718 showed that the presence of the MLG promoted a reduction in Ra and Rz, a better surface morphology. With respect to Inconel 625, the use of MLG platelets did not show any benefit on finish, although a significant reduction in grinding force was observed.

On the other hand, when the same metal is treated with different cutting fluids, the surface integrity can be significantly different. Vasu et al. [29] carried out an investigation on using the TRIM E709 emulsifier with Al_2_O_3_ nanoparticles to reduce the heat generated at the grinding zone for EN-31 steel. Results show that surface roughness and heat penetration were decreased with the addition of Al_2_O_3_ nanoparticles compared with dry and only TRIM E709 emulsifier. Yan et al. [30] focused on the effect of cutting fluid composition on the material deformation processes and experimentally investigated the influence of two cutting fluids (denoted as Blasocut and E709) on the high strain rate dynamic mechanical properties of a Ni-based superalloy, which was verified by the cutting process. Resulting from the work, Split Hopkinson Pressure Bar tests were consistent with those from cutting experiments, and the Rehbinder effect under E709 conditions was more significant, and the stress fluctuation was the largest, which is up to 14%.

The Rehbinder effect refers to the reduction of material hardness or ductility caused by the adsorption and reaction of a surfactant molecular film, which was first demonstrated by P.A. Rehbinder [31]. The surface tension of a crystal surface can be reduced by introducing certain surfactants and a Gibbs–Langmuir layer is subsequently formed. Barlow [32] pointed out that the Rehbinder effect is related to the decrease in strength and hardness caused by the adsorption of surfactant molecules, which weakens the bonds between elements on the lattice surface. At present, researchers generally believe that the active components adsorbed on the surface are involved in the breaking and rearrangement of atomic bonds, and the energy compensation of the broken atomic bonds leads to the reduction of surface energy, thus affecting the surface strength and mechanical properties of materials [33].

It can be seen from the above that the cutting fluid causes different surface integrity changes on the mechanical parts, which have an important influence on their service properties, especially on their resistance to crack initiation and corrosion on the surface. In order to further improve service performance by enhancing the surface integrity of ultra-high-strength steel structural parts, this paper focuses on the influence of cutting fluid on surface integrity, such as the surface morphology, residual stress, microhardness, and microstructure, which play a crucial role in the service performance of the key high-load structural parts. By comparing and analyzing the cutting force and surface integrity of different cutting fluids, the effect of the fluids on the surface integrity and the mechanism of action were revealed. This work may provide a creative control strategy for cutting fluid for improving surface integrity, which can also be used to evaluate and develop new cutting fluids to improve service performance.

## 2. Experimental Procedures

### 2.1. Material

The raw material used in the research is low-alloy, ultra-high-strength steel 45CrNiMoVA, which is usually used as the key structure of aircraft, such as landing gear, solid fuel rocket engine housing, shaft, gear, bearing, die, pad block, etc. The heat treatment state of the material is annealed, and it is a mixture of equiaxed ferrite phase (BCC crystal structure) and pearlite phase. Its chemical composition and physical properties are shown in Table 1 and Table 2, respectively. The heat treatment state of the material is the annealing state, and it is a mixture of equiaxed ferrite phase (BCC crystal structure) and the pearlite phase.

### 2.2. Split Hopkinson Pressure Bar Test

In order to better reveal the cutting mechanism and material response under cutting fluid during cutting, the dynamic mechanical property of 45CrNiMoVA was obtained by the split Hopkinson pressure bar (SHPB) technique, in which the shear strain rate simulated the corresponding shear strain characteristic in the material cutting process [34]. The sample for the SHPB test was hat-shaped, as shown in Figure 1. The shear band of 100 μm was represented in red; its size was much larger than the average grain size (13.44 μm) and excludes the influence of the size effect. The hat-shaped samples before and after the SHPB test are shown in Figure 2.

Prior to the SHPB test, the sample was immersed in the corresponding cutting fluid (HY-103, TRIM E709, or Vasco 7000) for 30 min. Then, the SHPB test was carried out on the samples at impact velocity levels of 15 m/s. Each test was repeated three times.

### 2.3. Turning Experiments

In order to study the influence of three cutting fluids on machining properties and the surface after machining, a single-factor turning test was designed. The selected cutting turning speed was 140 m/min, the feed rate was 0.15 mm/r, and the cutting depth was 0.20 mm. These three cutting fluids were delivered to the cutting zone by the conventional flood coolant technique at a flow rate of 2500 L/h. By calculation, the shear strain rate of material removal was about 125,000 s^−1^.

Cylindrical bars of 45CrNiMoVA steel subjected to the experiment were 50 mm in diameter and 300 mm in length. The cutting tool was the carbide tool ISCAR DNMG 150604-F3P, the shape parameters of which are shown in Table 3. The supporting tool holder is MDJNL2020K-15. The orthogonal turning experiments were carried out on a CNC lathe (CINCINATI-HAWK TC150, Cincinnati Machine Tool International Co., Ltd., Cincinnati, OH, USA), with a maximum power of 11 kW and a maximum speed of 5000 rpm. The Swiss Kistler force measuring system was adopted to measure these three cutting forces, namely, radial force *Fx*, tangential force *Fy*, and axial force *Fz*, in real time. The three-component dynamometer Kistler (type 9257B) was connected to the 8-channel charge amplifier (type 5070), and the voltage signal was converted into a digital signal through an A/D data conversion board. Finally, the digital signal was post-processed by Dynoware software (type 2825D-02) to obtain the cutting force data, as shown in Figure 3.

### 2.4. Integrity Indicator

Because the cutting fluid had an anti-corrosion function, the residual cutting fluid was used to preserve the specimen after processing in the factory. However, some cutting fluids contained corrosive elements (such as Cl and S) that could cause damage to the anticorrosive effect. Therefore, in order to simulate the actual working conditions, the surface morphology of the specimens was evaluated after one month (30 days). Surface morphology and surface roughness (S_a_) were measured by the VK-X100 laser 3D scanning microscope (KEYENCE CO., Ltd., Osaka, Japan). The roughness of the machined surface of each cutting fluid was measured 3 times, and the workpiece was rotated 120° after each time. The average of the three measurements is the result of the test.

The residual stress on the machined surface was measured by X-ray diffraction (XRD) at X-350 A. The interatomic spacings for a given lattice plane were determined for stress-free and residually stressed material by the apparatus. Thus, the residual stress could be calculated from the difference in lattice spacing. Based on the current study, the residual stress measurements were carried out using the sin^2^Ψ method with a Cr-Kα radiation source at the [211] lattice planes of the α-Fe [35]. The average of three measurements is taken as the final result. The microhardness of the affected layer under each cutting fluid was measured by a digital Vickers hardness tester (VTD 402, Beijing Wowei Technology Co., Ltd., Beijing, China). During the measurement, the loading value was 500 gf, and the load was kept for 10 s. Three locations were taken on each sample surface for measurement, and the average value was taken. In addition, the nano-indenter instrument (G200) (KLA Corporation, Milpitas, CA, USA) was used to test the samples, with a triangular pyramidal Berkovich indenter. During the measurement, the constant strain rate was 0.05 s^−1^ loading and the pressing depth was 100 nm. Each sample was tested continuously at 10 points along the layer depth from the surface of 0.6 μm at a distance of 4 μm. The average value was taken three times at different positions in the same depth.

Microstructure characterization of a machined surface under three cutting fluids requires the following processing to be obtained by Electron Backscattered Diffraction (EBSD): First of all, the sample was cut by WEDM (Wire Electrical Discharge Machining) for measurement, and the cross-section surface was polished successively from 200# to 3000# with SiC sandpaper until the surface was bright and clean without obvious scratches. Then the surface was electropolished with 7% HClO_4_ + 93% C_2_H_5_OH electrolytes at 450 mA/cm^2^ current density for 1 min. Finally, the EDAX Velocity Super Probe was applied for synchronous observation with EDAX EDS (Energy Dispersive Spectroscopy, Ametek Co., Ltd., San Diego, CA, USA), with 30.0 kV accelerating voltage and 0.1 μm step size. Accurate metallographic analysis of the acquired data was achieved by Chi-Scan TM (Tunneling Microscopy, Ametek Co., Ltd., San Diego, CA, USA) analysis technology.

## 3. Results

### 3.1. Physical and Chemical Properties

The physical and chemical properties of the material are the basis of the analysis, and here are measurements of ultra-high-strength steel and cutting fluid. The IPF (Inverse Pole Figure) mapping and grain size distribution of the material substrate are as shown in Figure 4. The average grain size is 13.44 μm.

The type of cutting fluid HY-103 (Yantai Runhua Chemical Products Co., Ltd., Yantai, China) is the cutting fluid employed in the investigated factory as a follow-up standard of comparison. The other two types of cutting fluids selected are TRIM E709 (Master Chemical Corporation, Perrysburg, OH, USA) and Vasco 7000 (Blaser Swisslube AG, Rüegsau, Switzerland). These three cutting fluids are applied at a dilution of 1:19 in deionized water. In terms of type, HY-103 is a mineral-based semi-synthetic micro-emulsion, TRIM E709 is a vegetable-based emulsion, and Vasco 7000 is a vegetable-based semi-synthetic micro-emulsion, the main additive compositions of which are shown in Table 4. Cl and S elements in cutting fluids are easy to cause corrosion to steel, and their contents are different, as shown in Table 5. The friction coefficient was measured at 300 °C by the SRV4 Wear Testing Machine. The cooling rate was measured by a thermocouple according to ISO 9950:1995 [36]. Surface tension and pH were also measured, and these test results are shown in Figure 5, Figure 6, Figure 7 and Figure 8.

### 3.2. Dynamic Mechanical Property

Through the SHPB test at 15 m/s, the shear strain rate under three cutting fluids is confirmed to be around 125,000 s^−1^, as shown in Figure 9, which is consistent with the shear strain rate in the turning experiment. Although these shear strain rates are similar, the failure stress (the maximum shear stress) of materials changes significantly under the influence of cutting fluids. As shown in Figure 10, the failure stresses of TRIM E709 and Vasco 7000 are reduced by 6.25% and 2.98%, respectively, compared to HY-103. After three repeated experiments, this trend is very stable. According to the research of Yan et al., the small effect of cutting fluid on dynamic mechanical properties will be significantly reflected in the cutting force [27].

### 3.3. Cutting Force

Three-direction cutting forces under HY-103 during cutting are shown in Figure 11. The mean results of the measured three-direction cutting forces under three cutting fluids are shown in Figure 12.

As shown in Figure 12, the cutting forces in three directions are HY-103 > Vasco 7000 > TRIM E709. In terms of *Fx*, TRIM E709 and Vasco 7000 reduce by 28.85% and 16.22%, respectively, compared to HY-103. In terms of *Fy*, TRIM E709 and Vasco 7000 decrease 34.74% and 15.28%, respectively, relative to HY-103. In terms of *Fz*, TRIM E709 and Vasco 7000 reduced by 52.57% and 27.57%, respectively, compared to HY-103. In the following discussion section, the action mechanism of cutting fluid on cutting force will be analyzed.

### 3.4. Surface Morphology

The material surface micromorphology under three cutting fluids after cutting is shown in Figure 13.

As shown in Figure 13, the periodic undulating groove texture formed by turning can be found on the surface, with no difference between these three cutting fluids for surface finish. The machining surface roughness (S_a_) measured 3 times under each cutting fluid is shown in Table 6 and Figure 14.

In terms of surface roughness S_a_, TRIM E709 is reduced by 15.7% and Vasco 7000 is reduced by 4.3% relative to HY-103, the trend of which is consistent with cutting force.

### 3.5. Residual Stress

The measurement results of surface residual stress measured 3 times under each cutting fluid are shown in Table 7 and Table 8 and Figure 15.

Under three cutting fluids, these residual stresses in both directions are tensile stresses. Compared with HY-103, TRIM E709 increases by 6.10% and Vasco 7000 decreased by 33.62% in the tangent direction (along the primary motion), and TRIM E709 and Vasco 7000 decreased by 4.45% and 7.60% relative to HY-103 in the axial direction (along the feed motion).

### 3.6. Microhardness

The microhardness of the machining surface measured 3 times under each cutting fluid and polished substrate plane (indicated by a dotted line) is shown in Table 9 and Figure 16.

As shown in Figure 16, the work hardening effect of HY-103 and Vasco 7000 is significant, and TRIM E709 decreases by 21.97% compared with HY-103. Vasco 7000 decreases by 0.73% compared with HY-103, with little change. Considering that the angle between the two opposite surfaces at the top of the diamond indenter is 136° and the diagonal length of indentations is 55~80 μm, the indentation depth after the calculation is 15.71~22.90 μm, which may cause the error due to pressing into the substrate. Therefore, a nano-indenter instrument is required to measure it along the cross-section radius. Figure 17 shows the distribution of nanohardness with depth.

The surface nanohardness trend in Figure 17 is consistent with the Vickers hardness result, in which the HY-103 and Vasco 7000 work hardening effects are significant and similar in degree. In addition, all of the surface nanohardness within 22.90 μm increased under three cutting fluids, indicating that the Vickers hardness results are accurate and reliable.

### 3.7. Microstructure Characterization

It is well known that macroscopic plastic deformation is the result of dislocation accumulation and its fluidity at the microscopic level. Under external mechanical and thermal loads, dislocations tangle and dislocation cells are formed with more intense plastic deformation [37]. Eventually, the original grains are segmented into the substructure on a sub-sized or nano-sized scale by the dislocation cell walls, which can be visually represented by EBSD technology. The grain boundary, misorientation angle distribution, and IPF mapping of machining surfaces under three cutting fluids (HY-103, TRIM E709, and Vasco 7000) are shown in Figure 18, Figure 19 and Figure 20. The red line and black line indicate high-angle grain boundaries (HAGBs, θ ≥ 15°) and low-angle grain boundaries (LAGBs, 2° < θ < 15°) in Figure 18. White arrows indicate the direction of tool movement in Figure 20.

From Figure 18, the depth of the grain refinement layers under these three cutting fluids is 9 μm, 4 μm, and 8 μm, respectively. TRIM E709 and Vasco 7000 decrease by 55.56% and 11.11% compared to HY-103, respectively. Figure 19 shows that the low-angle grain boundary frequencies under three cutting fluids are 42.01%, 49.23%, and 42.18%.

These grain size distributions within 10 μm on the surface processed by these three cutting fluids are as follows in Figure 21.

The average grain sizes of the surface layer under these three cutting fluids and material substrates are 0.12 μm, 0.16 μm, 0.20 μm, and 13.44 μm, respectively. These three cutting fluids all have a significant grain refinement effect on the material substrate shown in Figure 4, the mechanisms of which will be explained in detail in the discussion section.

## 4. Discussion

### 4.1. Dynamic Mechanical Property

It can be seen from Figure 10 that these three cutting fluids change the material failure stress (maximum shear stress) to varying degrees, i.e., HY-103 > Vasco 7000 > TRIM E709. The Rehbinder effect of cutting fluid can explain the intriguing phenomenon; however, there are few studies on it. Crystal breakage and a reduction in hardness were obviously observed when surfactants were added to metals, which were first reported by Rehbinder in 1928. Following that, the effect was found in polycrystalline materials to varying degrees [33].

Influenced by the Rehbinder effect, different surfactants in three cutting fluids penetrate into the shear zone (i.e., shear band during the test) along the surface microcracks, resulting in decreased surface energy and failure stress at different levels [30]. As can be seen from Table 4, TRIM E709 and Vasco 7000 have similar surfactant contents and are slightly higher than HY-103. Therefore, the failure stress under TRIM E709 and Vasco 7000 was reduced by 6.25% and 2.98%, respectively, compared to HY-103.

### 4.2. Cutting Force

The influence of cutting fluid on cutting force is multifaceted and complex. The response mechanism of cutting force is analyzed from the following aspects: lubrication performance, cooling rate, and the Rehbinder effect of cutting fluid.

Firstly, the lubrication performance of cutting fluid is very important to reduce cutting force, which is mainly related to lubricants and extreme pressure agents. In terms of lubricant content in Table 4, TRIM E709 (53.0~57.0, wt.%) is higher than HY-103 and Vasco 7000 (9.5~14 and 5.0~6.0, wt.%), thus providing the best lubrication performance.

As shown in Table 4, HY-103 has no extreme pressure agent, and the extreme pressure agent content of TRIM E709 (0.3~0.5, wt.%) is lower than that of Vasco 7000 (12.0~14.0, wt.%), resulting in the friction coefficient trend of HY-103 > TRIM E709 > Vasco 7000 at 300 °C in Figure 5. Considering the effects of the lubricant, the lubrication performance of E709 will be further improved, resulting in the cutting force trend of HY-103 > Vasco 7000 > TRIM E709 in Figure 11. In the heavy cutting process, the friction between the cutter and the chip will cause adhesive wear due to the high-temperature and high-pressure environment. For example, when the temperature of the deformation zone reaches more than 180 °C, the oily agent’s chemical adsorption film will be decomposed and lose its effect, so an extreme pressure agent must be used in the cutting fluid at this time [38]. Extreme pressure agents are mainly of three types: sulfur, chlorine, and phosphorus. The extreme pressure agent of TRIM E709 is organic acid phospholipids (0.3~0.5, wt.%), while that of Vasco 7000 contains vulcanized rapeseed oil (10.0~11.0, wt.%) and organic acid phospholipids (2.0~3.0, wt.%). The chemical reaction film is formed by a chemical reaction between sulfur, chlorine, phosphorus, and the metal surface at high temperatures. For example, sulfur atoms in vulcanized rapeseed oil react with metal to generate porous metal sulfides such as FeS on the metal surface. On the one hand, this compound can accommodate lubricants and form physical adsorption films more effectively. On the other hand, this strong compound film has a lower melting point than the metal, and when the metal is heated by pressure and friction, the film melts to produce a smooth surface, reducing the coefficient of friction and cutting force. These film-forming and film-melting temperatures for these compounds are shown in Table 10.

Secondly, a higher temperature can soften the material around the tool tip and thus reduce the cutting forces [39,40]. As shown in Figure 6, the cooling rate of cutting fluids is HY-103 > Vasco 7000 > TRIM E709 when the cutting temperature is between 300 °C and 750 °C in general. That is because TRIM E709 contains the maximum lubricant, which is oily in general and has poor cooling performance. Thus, the cutting force of the TRIM E709 is the lowest, resulting from the worst cooling rate and the greatest thermal softening effect, followed by the Vasco 7000.

Finally, influenced by the Rehbinder effect, the failure stress (maximum shear stress) of materials under TRIM E709 and Vasco 7000 reduces by 6.25% and 2.98% relative to HY-103 in Figure 10, which is conducive to reducing the cutting force during cutting. Thus, the cutting force of TRIM E709 is further reduced, followed by Vasco 7000, which is partly responsible for the trend of cutting force in Figure 12.

To sum up, lubricant, extreme pressure agent, cooling rate, and the Rehbinder effect of cutting fluid all have influence on cutting forces, and the trend is consistent, namely HY-103 > Vasco 7000 > TRIM E709. Cutting force is one of the most important cutting performance indexes in the process of cutting, which deeply affects the surface integrity. Therefore, in order to analyze the formation mechanism of surface integrity, the following analysis will first consider the influence of cutting fluids on cutting force and then the influence of the composition and performance of cutting fluid on integrity.

### 4.3. Surface Morphology

The increase in cutting force directly leads to severe plastic deformation and material flow on the machined surface and intensifies the undulating degree of texture, so the surface roughness (S_a_) is higher. Thus, the trend of S_a_ in Figure 14 is consistent with that of cutting force in Figure 12.

In terms of the surface finish caused by the flank face of the tool, there is no obvious difference under three cutting fluids. There are two reasons for this. On the one hand, the lower surface tension of the cutting fluid makes it favorable to penetrate into the area between the flank face and machining surface [41]. The molecular structure of surfactants is amphoteric: one end is a hydrophilic group, such as carboxylic acid, sulfonic acid, sulfuric acid, amino or amine groups, and their salts; the other end is a hydrophobic group, such as hydrocarbon chains with more than 8 carbon atoms. Thus, surfactants are effective in reducing the surface tension of the cutting fluid [42]. Due to the small difference in surfactant content among the three cutting fluids, there is no significant difference in surface tension, with the trend of TRIM E709 > HY-103 > Vasco 7000 in Table 4 and Figure 7. On the other hand, although the three cutting fluids contain different contents of lubricant and extreme pressure agent shown in Table 4, the friction coefficient shown in Figure 5 is too small to cause the difference in surface finish.

As mentioned in Section 2.1, Cl and S in the cutting fluid can easily corrode metals, especially on a freshly machined surface. However, it is difficult to observe corrosion products by a low magnification microscope due to the action of corrosion inhibitors. Figure 22 shows the corrosion morphology under three cutting fluids by a scanning electron microscope (SEM). Energy dispersive spectroscopic (EDS) analysis of pitting products on the machined surface is shown in Figure 23.

From the above figure, we can see the corrosion degree of three cutting fluids (HY-103 > TRIM E709 > Vasco 7000) and the content of C and O of pitting corrosion increasing relative to the raw material 45CrNiMoVA. Pitting products occur in two types (spherical and floccule, similar in composition), and they mostly occur at the valley of the topography groove, where conditions are conducive to the accumulation of corrosive media. According to the different corrosion mechanisms of metals, they can be divided into chemical corrosion and electrochemical corrosion, and electrochemical corrosion is in the majority. With the migration of electrons and ions, various galvanic cells form in the corrosion zone [43]. Pitting corrosion is mainly due to the following chemical reactions:

The anode: Fe-2e → Fe^2+^

The cathode: 2H_2_O + 2e → H_2_↑ + 2OH (pH < 7)/O_2_ + 2H_2_O + 4e → 4OH^−^ (pH ≥ 7)

Dissolution of Fe occurs at the anode, and oxygen absorption corrosion or hydrogen evolutional corrosion occurs at the cathode. According to Figure 8, the pH of all three cutting fluids is greater than 9, so only oxygen-absorbing corrosion occurs. As the concentration of O_2_ in the valley is lower than that at the peak, a potential gradient is formed, which further accelerates the corrosion. Subsequently, Fe^2+^ and OH^−^ will continue to undergo a series of electrochemical changes in the migration process, forming stable corrosion products Fe_2_O_3_ and FeCO_3_, which can protect the metal from corrosion [44]. However, Cl^−^ penetrates these protective films to generate β-FeOOH and Fe_6_(OH)_12_CO_3_, and S promotes the formation of α-FeOOH, which destroys the protective film and causes the corrosive medium to penetrate into the internal material to start a new corrosion cycle [45]. The synergistic effect of S and Cl will promote the corrosion of steel, making the corrosion rate greater than that of a single element [46].

A corrosion inhibitor is necessary to delay the above corrosion by creating a stronger protective film. According to the type of protective film formed, corrosion inhibitors can be divided into oxidation film, deposition film, and adsorption film corrosion inhibitors [47]. It can be seen from Table 5 that HY-103 contains Cl (921.19 mg/L) and S (908.92 mg/L), TRIM E709 only contains Cl (1023.9 mg/L), and Vasco 7000 only contains a small amount of S (49.14 mg/L). Contrary to this trend, the content of corrosion inhibitor is highest in Vasco 7000 (11.4~14.6, wt.%), followed by TRIM E709 (8.1~11.1, wt.%), and HY-103 (3.3~4.5, wt.%), as shown in Table 4. Therefore, HY-103 has the highest corrosion and Vasco 7000 has the best corrosion resistance.

### 4.4. Residual Stress

As is known, three factors mainly cause surface residual stress: mechanical effect, thermal effect, and phase change [48]. Due to the cooling effect of the cutting fluid, the cutting temperature does not reach the phase transition temperature [40]. Therefore, we mainly consider the residual stress caused by mechanical and thermal effects.

The mechanical effect is divided into two kinds: one is the compressive plastic deformation (squeeze effect) caused by the *Fx* on the surface material, and the elastic recovery of the internal material produces residual compressive stress in tangential and axial directions after cutting. The other is the tensile plastic deformation (stretching effect) caused by *Fy* and *Fz* on the surface material; the elastic recovery of the internal material produces residual tensile stress in tangential and axial directions after cutting. In terms of mechanical effects caused by cutting forces, as the stretching effect is greater than the squeeze effect, the tangential and axial tensile stresses are consistent with the cutting force, i.e., HY-103 > Vasco 7000 > TRIM E709 in Figure 12.

The thermal effect refers to the heat generated by friction during the cutting process that makes the surface material expand. After cutting, the elastic recovery of the surface material is restrained by the internal material, resulting in residual tensile stress in tangential and axial directions. Based on the cooling rates shown in Figure 6, TRIM E709 has the lowest cooling performance and the highest cutting temperature, so the tensile stress under TRIM E709 increases more than that under the other two cutting fluids. In the tangent and axial directions, the trend is TRIM E709 > HY-103 > Vasco 7000 and HY-103 > TRIM E709 > Vasco 7000, as shown in Figure 15.

### 4.5. Microhardness

The hardening of the surface layer is caused by the cutting force in the mechanical processing. Under cutting force, the plastic deformation of surface material causes severe distortion, elongation, fibrosis, and breakage of the metal grains, thus hindering the further deformation of the metal, improving the surface hardness, and reducing the plasticity [40,49]. Therefore, the work hardening under these three cutting fluids shows the same trend as that of the cutting force.

### 4.6. Microstructure Characterization

In the process of cutting, HAGBs are usually formed at the grain boundary, as shown in Figure 18. These forces result in severe plastic deformation of the subsurface layer supporting grain refinement [50]. The higher the cutting force, the greater the plastic deformation, the smaller the low-angle grain boundary frequency, and the deeper the grain refinement depth [40]. Therefore, the trend of grain refinement layer depth in Figure 18 is consistent with the cutting force shown in Figure 12. This view is also verified by the change in low-angle grain boundary frequency in Figure 19. In addition, it is observed from the IPF mapping in Figure 20 that there is obvious material plastic flow along the cutting force direction on the surface layers of HY-103 and Vasco 7000, which is because these two cutting fluids cause higher cutting forces compared with TRIM E709 in Figure 12.

The above phenomenon can also be verified by the grain orientation difference, which is defined as the angle of rotation of a crystal lattice around the orientation difference axis shared by the crystal lattice and its neighbors. The orientation difference describes the relative crystallography difference of two grains, corresponding to the lattice distortion, indicating the degree of plastic deformation and dislocation accumulation. Kernel Average Misorientation (KAM) describes the mean orientation difference between the measurement point and the nearest third-order kernel (the default maximum orientation difference is 5°), which is a physical characterization of the lattice distortion degree and local plastic deformation of grain. The KAM of the cross section of the surface processed by the three cutting fluids is shown in Figure 24. As can be seen from the figure, relative to the grains far away from the machined surface, the local plastic deformation caused by the cutting force on the machined surface leads to significant local orientation distribution within the grains, which induces a large amount of dislocation density in the grains. Therefore, a small angle grain boundary zone with a certain depth is formed on the surface of the processing sub-surface. When the local stress exceeds the strength limit of the grain, the overloaded grain will be broken and refined. It can be seen in Figure 24a,c, that, compared with Figure 24b, grains beyond the analytical accuracy of EBSD appear near the surface, so they are shown as black spots. HY-103 cutting fluid has the largest cutting force and the most significant lattice distortion, while TRIM E709 cutting fluid has the smallest cutting force, the smallest plastic deformation, and the smallest lattice distortion layer depth.

The average grain size of the surface layer under TRIM E709 (0.16 μm) increases by 33.33% relative to HY-103 (0.12 μm), resulting from a lower cutting force under TRIM E709 than HY-103. Compared with HY-103 (0.12 μm), the average grain size under Vasco 7000 (0.20 μm) increases notably by 66.67%, due to the lower cutting force under Vasco 7000 than HY-103. Interestingly, cutting force under Vasco 7000 is higher than TRIM E709, as shown in Figure 12, and the average grain size under Vasco 7000 (0.20 μm) is larger than TRIM E709 (0.16 μm), as shown in Figure 16. This is because the grain is elongated by the cutting force but not broken, causing fibrosis and increasing grain size. According to the literature, although surface grain does not break after elongation, fibrosis appears, which also causes work hardening obviously [51,52].

Moreover, the growth of refined grain may occur due to local thermal effects in the cutting process, but a higher cooling rate of cutting fluid may inhibit the growth of refined grain due to heat conduction and convection near the surface [53]. In addition, surfactants adsorbed on grain boundary surfaces repel each other under the Rehbinder effect and inhibit the growth of grains. Thus, a gradient of grain refinement can be observed progressively deeper into the substrate material, while the sub-surface materials are usually composed of refined grains [54]. The phenomenon can obviously be observed in Figure 20 and Figure 21, and grain refinement and depth are different due to the composition and cooling rate of the cutting fluids. As shown in Table 4 and Figure 6, for the three cutting fluids, the difference in cooling rate is significantly greater than that of surfactant content, so the growth of refined grain inducted by the thermal effect plays a major role.

The polar figure (PF) mapping details the crystallographic texture orientation of the machined surface material in Figure 25. Machined surface material under TRIM E709 has a high density along the {001} direction with the greatest magnitude of 37.22 MUD, much higher than HY-103 (5.12 MUD) and Vasco 7000 (7.50 MUD). To more clearly represent the crystallographic orientation distribution, the IPF mappings of machining surfaces are shown in Figure 26. The Z0 IPF mapping of Figure 26b describes the highest texture index of 9.27 for machined surface materials under TRIM E709, higher than HY-103 (2.27 MUD) and Vasco 7000 (2.35 MUD), and indicates that most grains are inclined to the {001} direction. IPFs generally map along each of the three characteristic crystallographic directions, which is consistent with the interpretation of EBSD in Figure 20. It means the randomness of the microstructural directions of HY-103 and Vasco 7000 while the growth of refined grain under TRIM E709 along the {001} direction occurs. This is because the cooling rate of TRIM E709 is significantly lowest among these cutting fluids, leading to the growth of refined grain along {001} with higher temperature at deeper depths, resulting in shallower grain refinement layers, followed by Vasco 7000. This explains the tendency of grain refinement layers in Figure 18.

## 5. Conclusions

To improve the service performance of ultra-high-strength steel components, through the analysis of the composition and properties of three cutting fluids, the effect of cutting fluids on the surface integrity and the mechanism of action are revealed. These specific conclusions are as follows:Because of the different composition and properties of the three cutting fluids, the changing trend of the cutting forces in three directions is HY-103 > Vasco 7000 > TRIM E709;The TRIM E709 contains the maximum lubricants, which reduce cutting force and surface roughness (S_a_), while the Vasco 7000 contains the minimum corrosive elements, which results in the least pitting;Both tangential and axial stresses under cutting fluid are tensile stresses. TRIM E709 and Vasco 7000 are reduced axially by 4.45% and 7.60% relative to HY-103, respectively;The grain refinement layer depths of HY-103, TRIM E709, and Vasco 7000 are 9 μm, 4 μm, and 8 μm, respectively, and TRIM E709 can induce recrystallized grains to grow along {001}.

## Figures and Tables

**Figure 1 materials-16-03331-f001:**
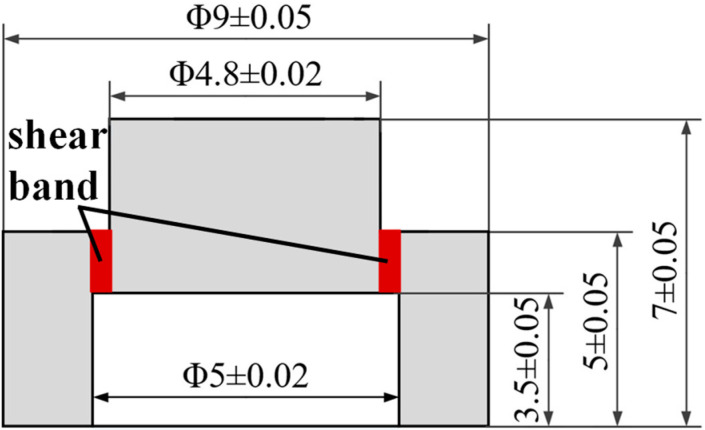
Size of the hat-shaped sample (mm).

**Figure 2 materials-16-03331-f002:**
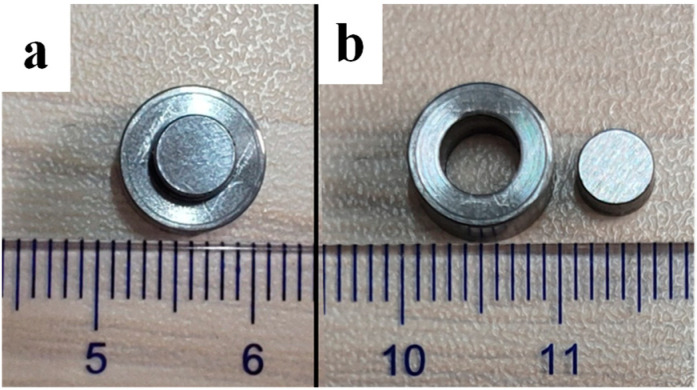
Hat-shaped samples: (**a**) before and (**b**) after the Hopkinson pressure bar tests (mm).

**Figure 3 materials-16-03331-f003:**
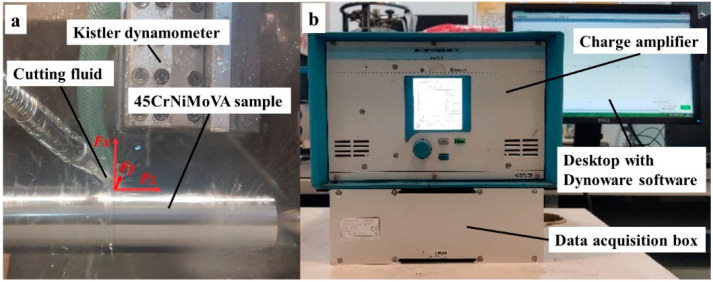
(**a**) 45CrNiMoVA workpiece in the cutting and (**b**) dynamometer set up for cutting force recording.

**Figure 4 materials-16-03331-f004:**
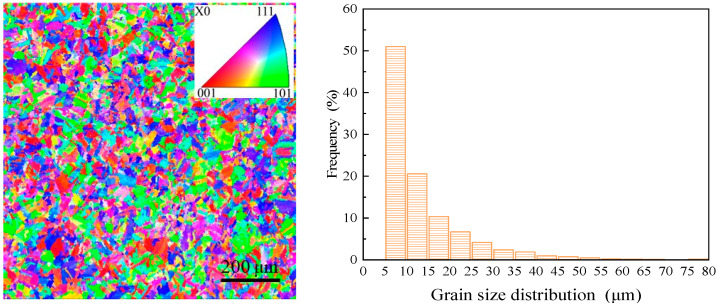
The IPF mapping and grain distributions of 45CrNiMoVA.

**Figure 5 materials-16-03331-f005:**
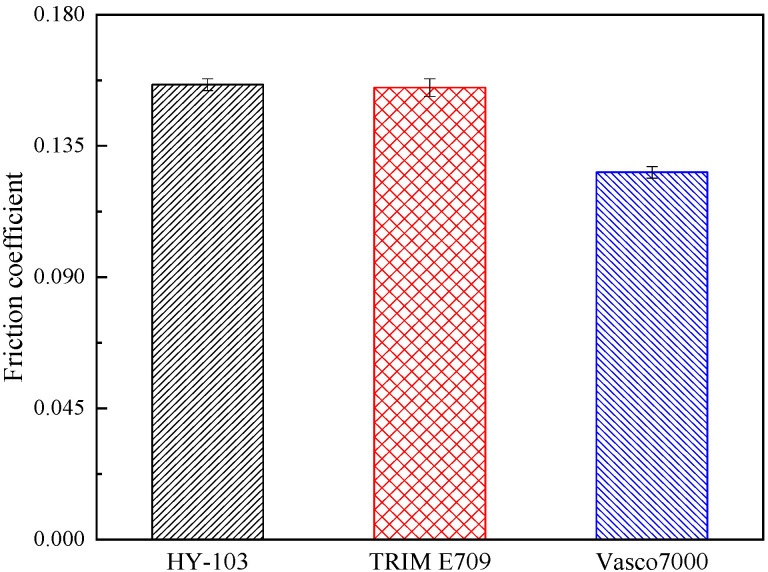
Friction coefficients of three cutting fluids.

**Figure 6 materials-16-03331-f006:**
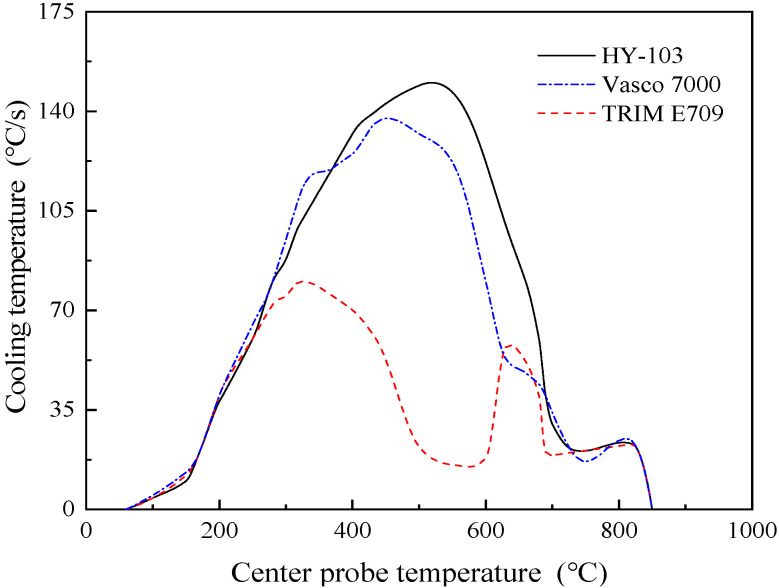
Cooling rates of three cutting fluids.

**Figure 7 materials-16-03331-f007:**
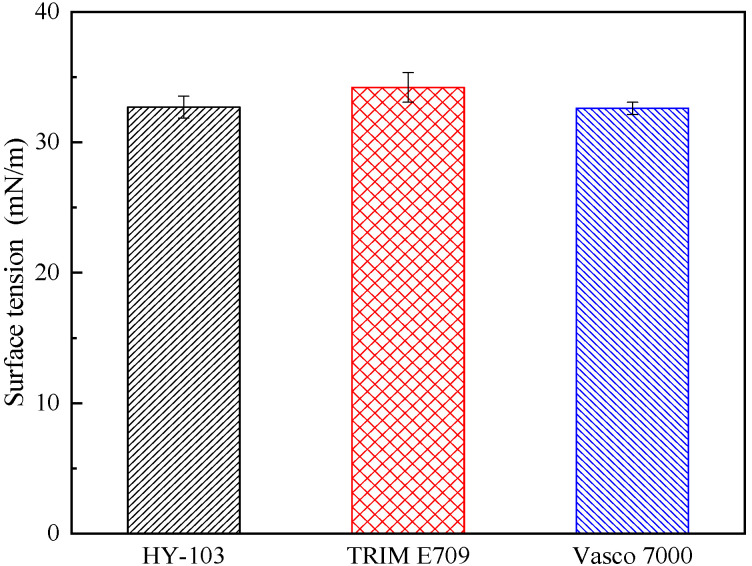
Surface tension of three cutting fluids.

**Figure 8 materials-16-03331-f008:**
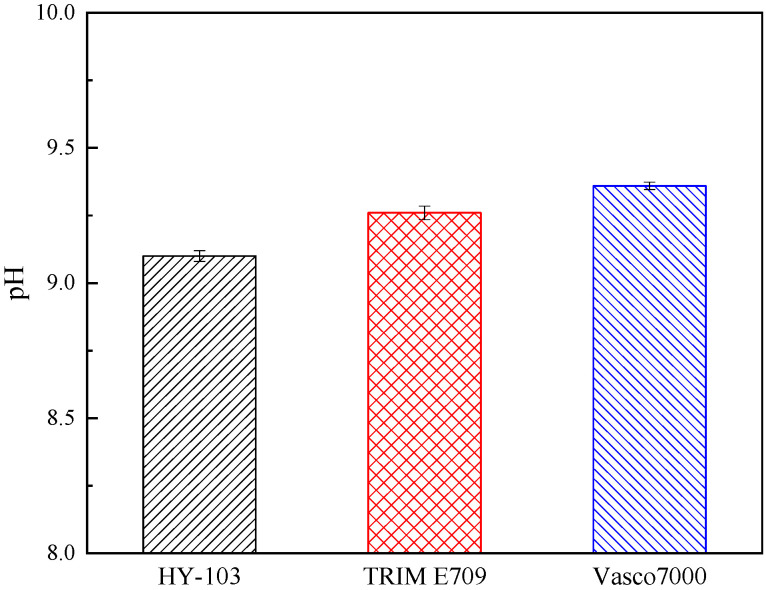
pH of three cutting fluids.

**Figure 9 materials-16-03331-f009:**
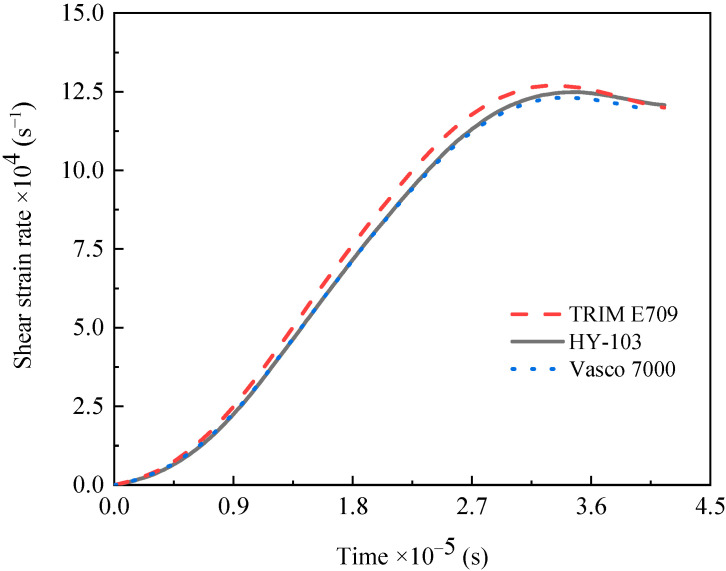
Shear strain rate of material under three cutting fluids.

**Figure 10 materials-16-03331-f010:**
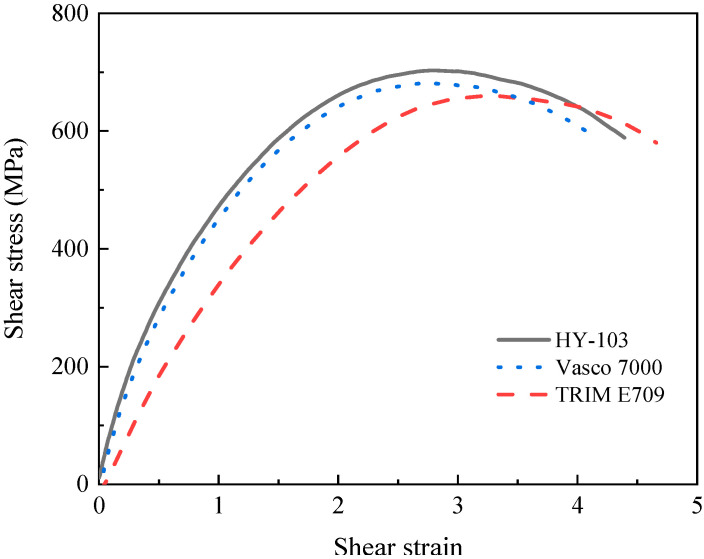
Shear stress-strain curve of material under three cutting fluids.

**Figure 11 materials-16-03331-f011:**
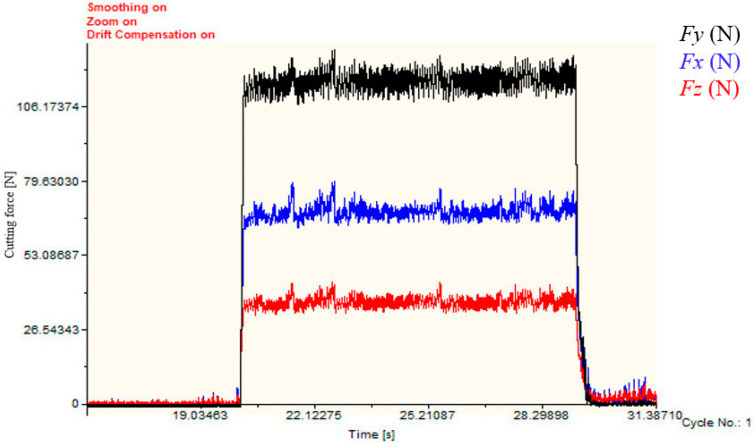
Three-direction cutting forces of 45CrNiMoVA under HY-103.

**Figure 12 materials-16-03331-f012:**
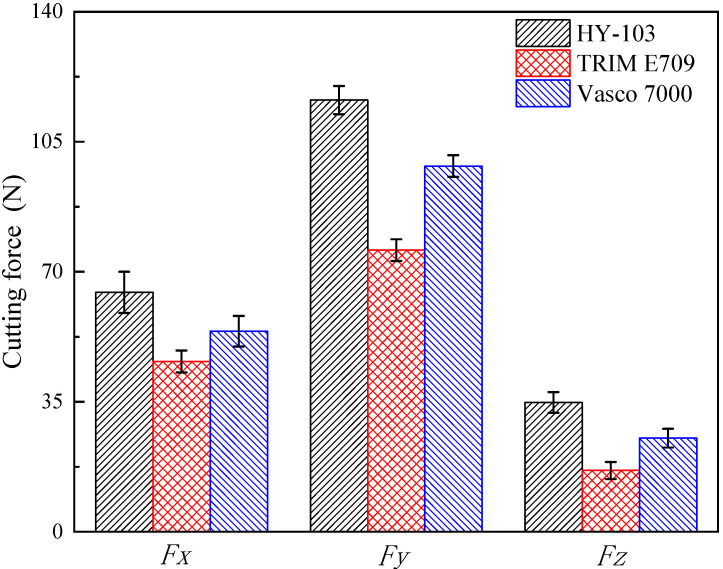
Three-direction cutting forces of 45CrNiMoVA under three cutting fluids.

**Figure 13 materials-16-03331-f013:**
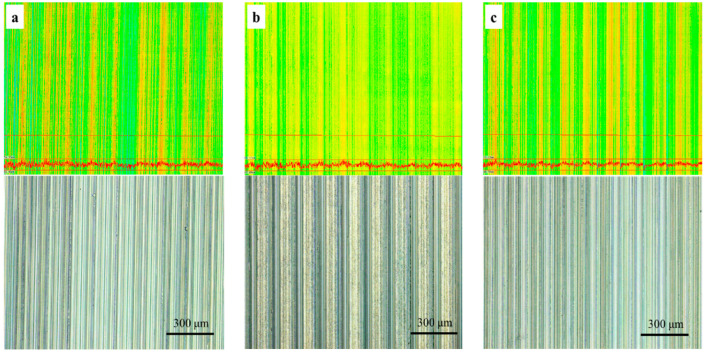
Material surface micromorphology under three cutting fluids: (**a**) HY-103, (**b**) TRIM E709, and (**c**) Vasco 7000.

**Figure 14 materials-16-03331-f014:**
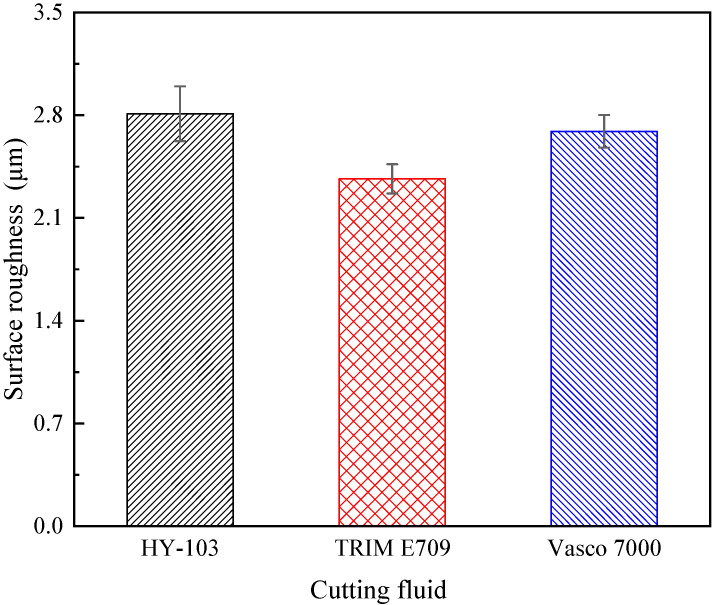
The machining surface roughness under three cutting fluids.

**Figure 15 materials-16-03331-f015:**
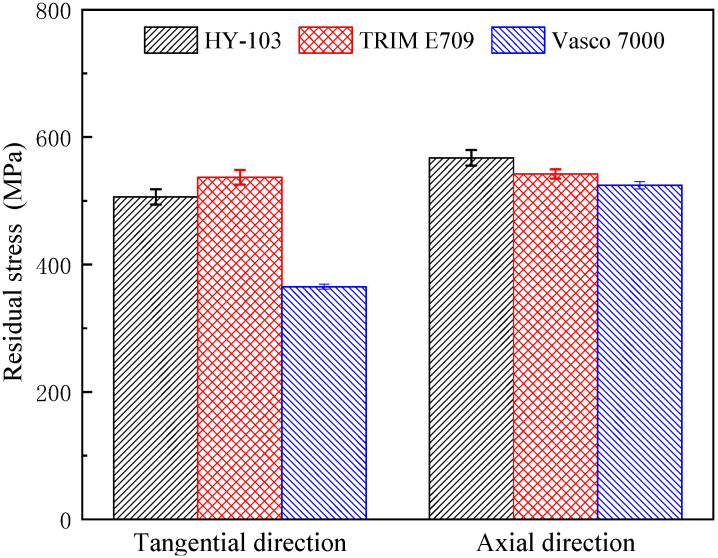
Material surface residual stresses under three cutting fluids.

**Figure 16 materials-16-03331-f016:**
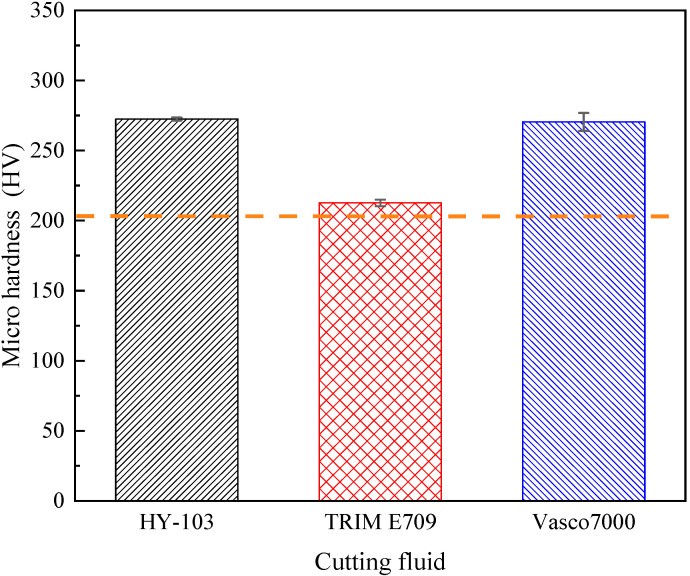
Microhardness of machining surface under cutting fluids and substrate polished surface.

**Figure 17 materials-16-03331-f017:**
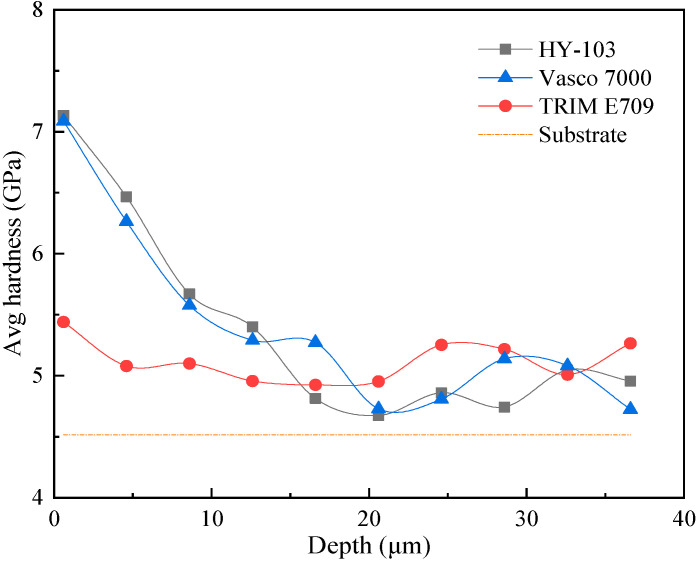
The distribution of nanohardness with depth.

**Figure 18 materials-16-03331-f018:**
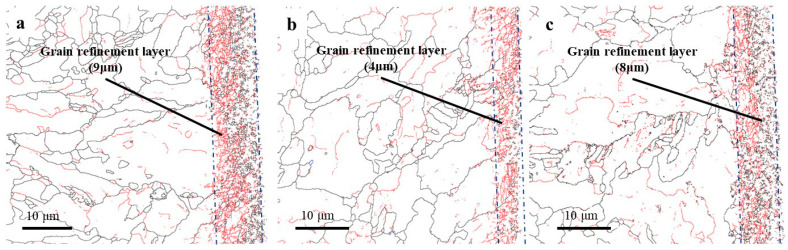
Cross-sectional grain boundary mappings of machining surfaces under three cutting fluids: (**a**) HY-103, (**b**) TRIM E709, and (**c**) Vasco 7000.

**Figure 19 materials-16-03331-f019:**
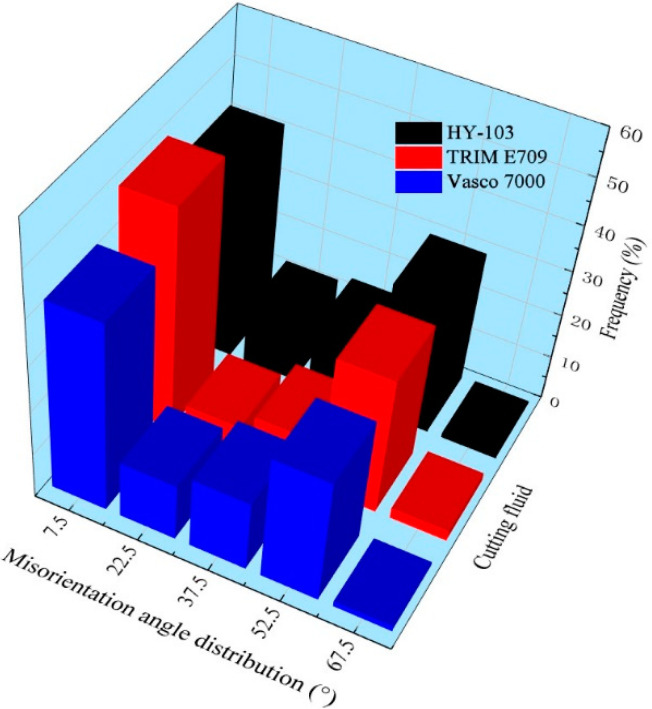
The misorientation angle distributions under these three cutting fluids.

**Figure 20 materials-16-03331-f020:**
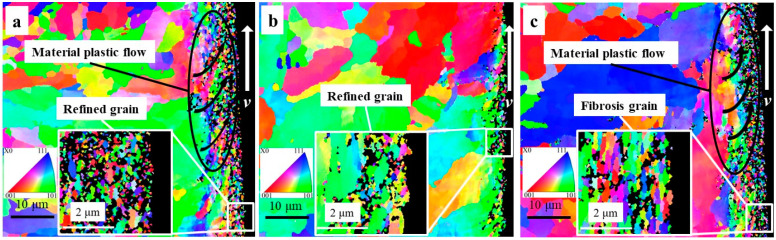
Cross-sectional IPF mappings of machining surfaces under three cutting fluids: (**a**) HY-103, (**b**) TRIM E709, and (**c**) Vasco 7000.

**Figure 21 materials-16-03331-f021:**
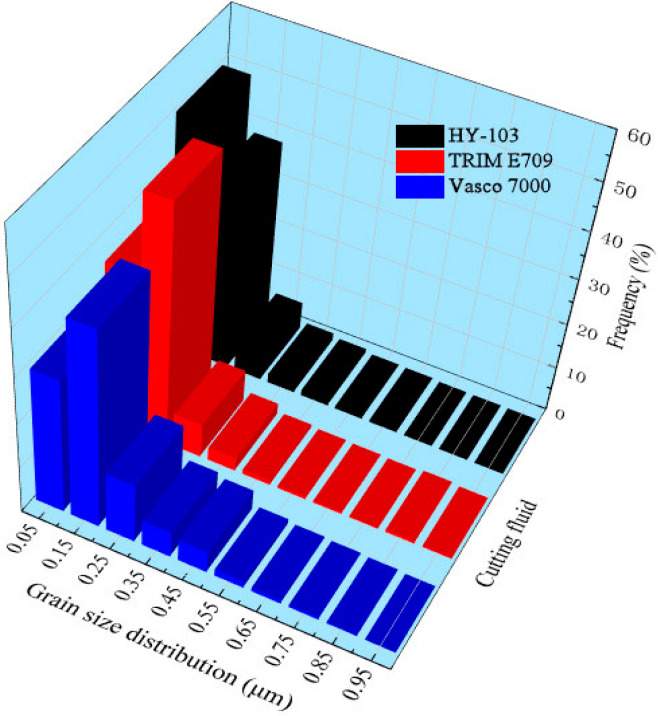
The grain distributions under these three cutting fluids.

**Figure 22 materials-16-03331-f022:**
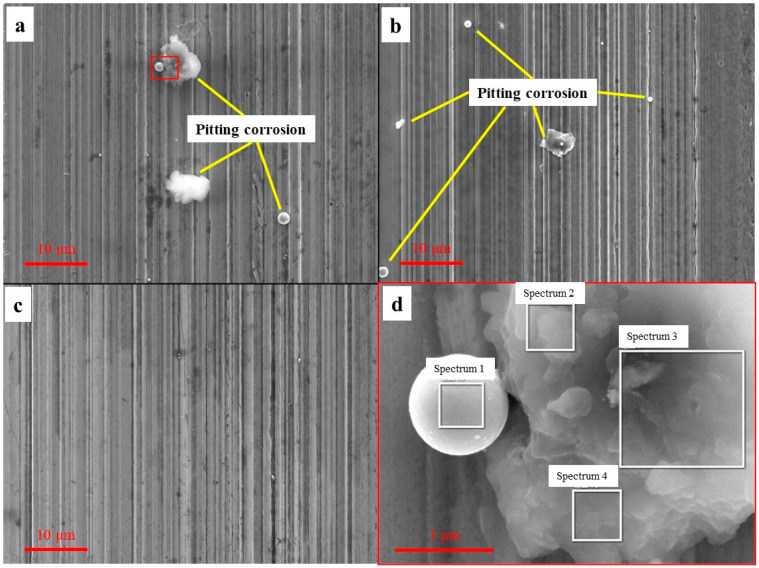
SEM images of the machined surfaces under three cutting fluids: (**a**) HY-103, (**b**) TRIM E709, and (**c**) Vasco 7000; (**d**) enlarged view of the red box area in (**a**).

**Figure 23 materials-16-03331-f023:**
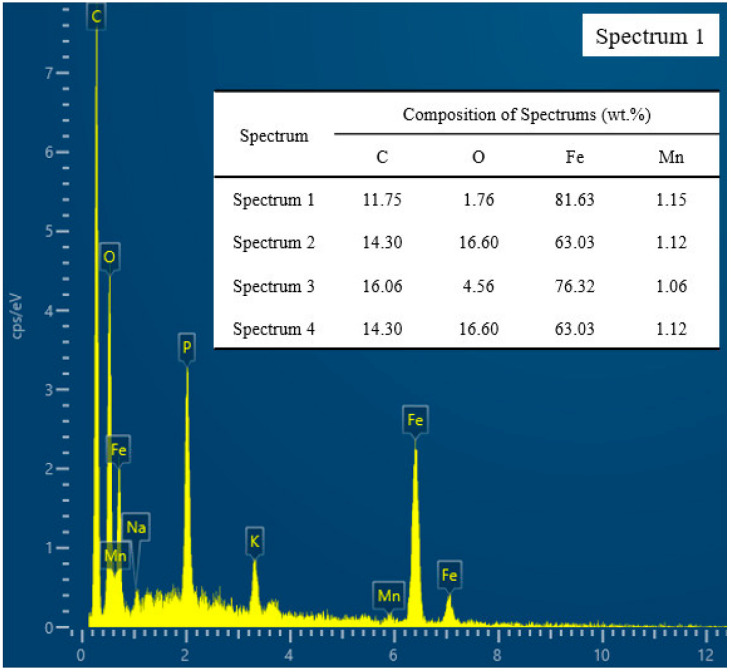
EDS analysis of pitting products on the machined surface.

**Figure 24 materials-16-03331-f024:**
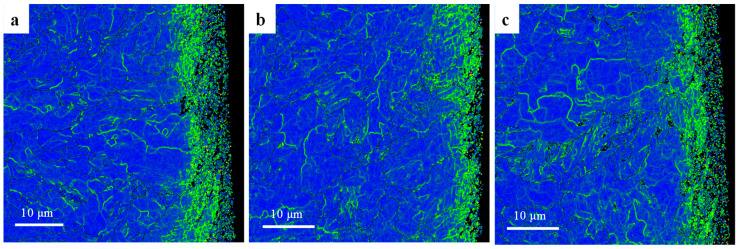
Cross-sectional KAM mappings of machining surfaces under three cutting fluids: (**a**) HY-103, (**b**) TRIM E709, and (**c**) Vasco 7000.

**Figure 25 materials-16-03331-f025:**
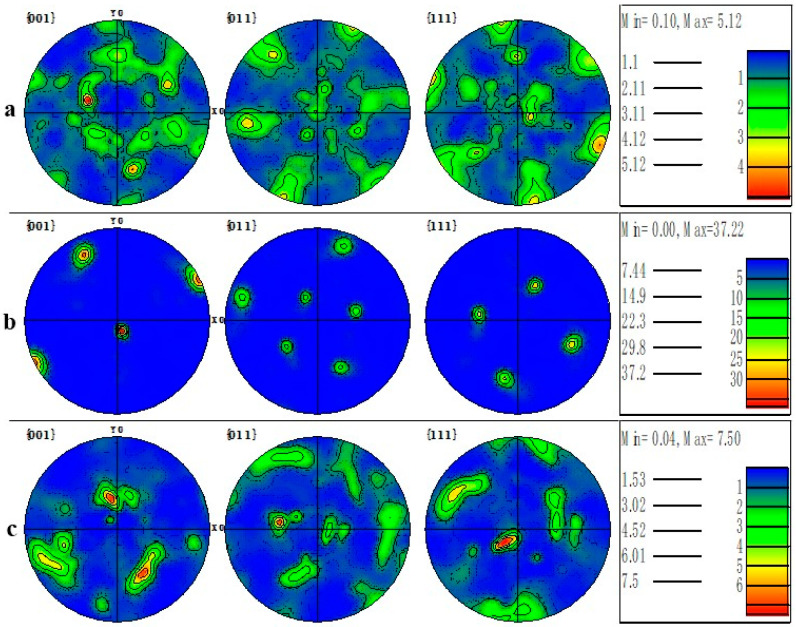
Cross-sectional PF mappings of the machined surface under three cutting fluids: (**a**) HY-103, (**b**) TRIM E709, and (**c**) Vasco 7000.

**Figure 26 materials-16-03331-f026:**
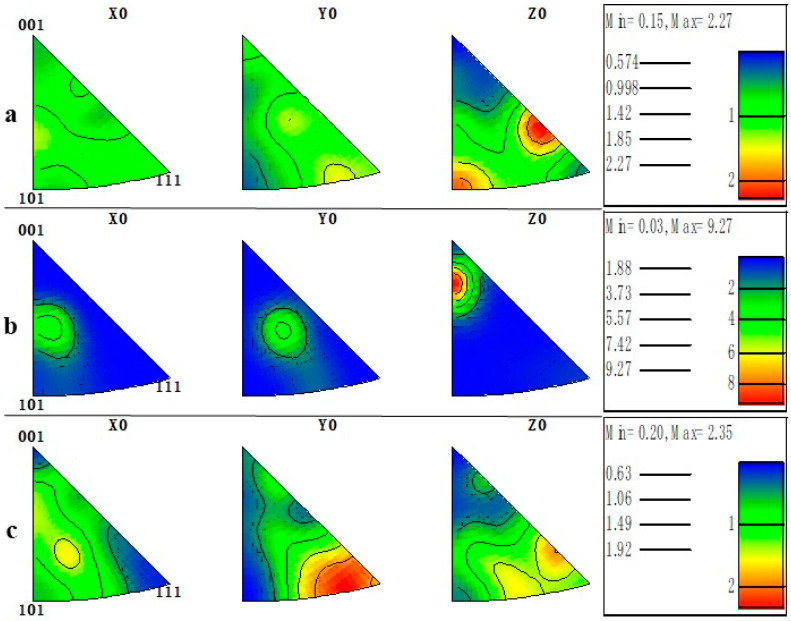
Cross-sectional IPF mappings of machining surfaces under three cutting fluids: (**a**) HY-103, (**b**) TRIM E709, and (**c**) Vasco 7000.

**Table 1 materials-16-03331-t001:** The main chemical composition of 45CrNiMoVA.

Element	Composition (wt.%)	Element	Composition (wt.%)
Ni	1.30~1.80	Si	0.17~0.37
Cr	0.80~1.10	Mn	0.50~0.80
V	0.10~0.20	C	0.42~0.49
Mo	0.20~0.30	Cu	≤0.025
S	≤0.025	P	≤0.025
Fe	Base		

**Table 2 materials-16-03331-t002:** The main physical properties of annealed 45CrNiMoVA [1].

Young’s Modulus/GPa	Yield Stress/MPa	Tensile Strength/MPa	Elongation/%
215	512	892	51

**Table 3 materials-16-03331-t003:** The shape parameters of the cutting tool.

Rake Angle $\gamma_{0}$ (°)	Relief Angle $\alpha_{0}$ (°)	Tool Cutting Edge Angle $\kappa_{r}$ (°)	Tip Radius $r_{\varepsilon }$ (mm)
12.5	0	90	0.4

**Table 4 materials-16-03331-t004:** The main additive composition of cutting fluids.

Additive	Composition of Cutting Fluids (wt.%)		
	HY-103	TRIM E709	Vasco 7000
pH regulator	6.0~8.0	8.1~11.2	21.0~24.0
Lubricant	9.5~14	53.0~57.0	5.0~6.0
Corrosion inhibitor	3.3~4.5	8.1~11.1	11.4~14.6
Emulsifier	3.0~4.0	4.0~5.0	7.2~9.4
Surfactant	20.0~28.0	23.0~32.0	22.1~26.7
Defoaming agent	0.1~0.2	0.1~0.2	0.1~0.2
Extreme pressure agent	0.0	0.3~0.5	12.0~14.0
Other	Last	Last	Last

**Table 5 materials-16-03331-t005:** The main corrosive element content of cutting fluids.

Element	Content of Cutting Fluids (mg/L)		
	HY-103	TRIM E709	Vasco 7000
Cl	921.19	1023.9	0
S	908.92	0	49.14

**Table 6 materials-16-03331-t006:** The statistical value of surface roughness under three cutting fluids.

Statistical Categories	Surface Roughness (μm)		
	HY-103	TRIM E709	Vasco 7000
Average	2.81	2.37	2.69
Uncertainty (*p* = 0.95)	0.46	0.25	0.27
Standard deviation	0.19	0.10	0.11

**Table 7 materials-16-03331-t007:** The statistical value of tangent surface residual stresses under cutting fluids.

Statistical Categories	Residual Stresses (MPa)		
	HY-103	TRIM E709	Vasco 7000
Average	506.10	536.97	365.27
Uncertainty (*p* = 0.95)	30.05	28.57	9.69
Standard deviation	12.10	11.50	3.90

**Table 8 materials-16-03331-t008:** The statistical value of axial surface residual stresses under cutting fluids.

Statistical Categories	Residual Stresses (MPa)		
	HY-103	TRIM E709	Vasco 7000
Average	567.47	542.17	524.30
Uncertainty (*p* = 0.95)	30.50	18.01	13.99
Standard deviation	12.28	7.25	5.63

**Table 9 materials-16-03331-t009:** The statistical value of microhardness of machining surface under cutting fluids.

Statistical Categories	Microhardness (HV)		
	HY-103	TRIM E709	Vasco 7000
Average	272.42	212.57	270.43
Uncertainty (*p* = 0.95)	2.88	5.87	1.29
Standard deviation	1.15	2.36	6.47

**Table 10 materials-16-03331-t010:** Film-forming and film-melting temperatures of extreme pressure agents.

Extreme Pressure Agent	Extreme Pressure Film	Film-Forming Temperature/°C	Film-Melting Temperature/°C
Chlorine	Metal chloride	180	670
Phosphorus	Metal phosphide	280	950
Sulfur	Metal sulfide	520	1100

## Data Availability

The data that support the findings of this study are not publicly available due to confidentiality restrictions.

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
