# Peer review of "The Effect of Cutting Fluid on Machined Surface Integrity of Ultra-High-Strength Steel 45CrNiMoVA"

_materials, 2023, doi:10.3390/ma16093331_

Round 1
Reviewer 1 Report
Please address all comments in the attached file. Besides, improve the discussion in the text and fix the errors in the text.

English quality is fairly good.
Reviewer 2 Report
Paper is in major revision quality level, more or less.
This paper discusses a research study that investigated the effect of different cutting fluids on the surface integrity of ultra-high-strength steel. The study looked at three different cutting fluids - HY-103, TRIM E709, and Vasco 7000 - and examined various aspects of surface integrity, including cutting force, surface morphology, residual stress, microhardness, and microstructure.
The results of the study showed that the cutting forces in three directions followed the trend of HY-103 > Vasco 7000 > TRIM E709. TRIM E709 contained the maximum amount of lubricants, which reduced cutting force and roughness Sa. Vasco 7000 had the least amount of corrosive elements, resulting in the least pitting. Both tangential stress and axial stress under cutting fluid were tensile stress, with TRIM E709 and Vasco 7000 reducing axial stress by 4.45% and 7.60%, respectively, relative to HY-103.
On the other hand , all the work by Lopez de Lacalle, or O.Pereira about effect of cutting fluid on machined surface integrity are missed and we are referring to 10-15 recent ones, all in Inconel 718. In addition, you missed the new testing methods based on residual and surface integrity, see https://doi.org/10.1177/095440621561614. The new method is opening new ways in machining and coolant testing as well.
In the missed references you will see the new application in aero turbines.
The current state is difficult to review, many typos, and missed works, so reject is the only solution.
Figure 6 to 8 can be eliminated.
45CrNiMoVA, this is in line with Works like works about Superalloys, for instance,
Surface integrity and fatigue of non-conventional machined Alloy 718, Journal of
Manufacturing Processes 48, 44-50.
Coolants can be revised in term of physical characteristics, in Sustainability analysis of lubricant oils for minimum quantity lubrication based on their tribo-rheological performance, Journal of Cleaner Production 164, 1419-1429
Rewrite the paper; complete the state of the art with the new 20 references (more can be missed) and resubmit soon. Include the key reference works and make a better discussion.
Many typos of missed references.
Reviewer 3 Report
The Authors focused on the surface integrity of ultrahigh strength steel which has significant influence on service performance and cutting fluid that plays an important role on surface integrity in production. In this paper, the surface integrity of ultra-high-strength steel under three cutting fluids have been investigated from the aspects of cutting force, surface morphology, residual stress, micro hardness, and microstructure. The article seems to be interesting for scholars and engineers. It creates a logical scientific experimental research and in my opinion could be published in "Materials". Some of the comments on the manuscript are listed below.
1) Line 26 and 27; some keywords have been already used in the title of the manuscript. Please change them into different ones (to avoid the keywords repetition with the words used in the title).
2). Line 159; what does 0.15 mm/z mean?
English technical language is quite good.
Reviewer 4 Report
Not sure what do you mean by economical strategy as you don’t have studied the cost in this work
“Ultra-high-strength steel”..” of high-strength” do not have a proper flow
“environmental impact” what does means?
As above “high-strength steel is common to apply cutting fluid in the cutting process.” The English is very weak as the cutting fluid is applied for materials machined and not the other way
First paragraph almost do not have sense as English grammar
Not clear which is the link of citation [15] as there is not any relationship between this work and that research!
I advice to re do the introduction as it is incomplete and also misleading with the data. This is because there are plenty literature which are focused in this research aspect and were not considered and noted different references which do not have any link with present research
“f 45CrNiMoVA (industry standard).” Which one ?
“was observed after” better using “was evaluated”
“. The final measurement result was the average of all measurements” please reformulate
“The measurement method was consistent with the sur- 194 face morphology.” What do you mean here consistent ? this is not clear sentence
Figure 5 requires the standard deviation – the same figure 7 and 8, 12
“Under the external mechanical.. “ please provide evidence such as TEM measurement or a citation for this
I found the English quality very weak.
Round 2
Reviewer 2 Report
It is OK
Reviewer 4 Report
.
.